# Adaptive and freeze-tolerant heteronetwork organohydrogels with enhanced mechanical stability over a wide temperature range

Hainan Gao[1,2], Ziguang Zhao[1], Yudong Cai[2], Jiajia Zhou[1,3], Wenda Hua[4], Lie Chen[1], Li Wang[2], Jianqi Zhang[4], Dong Han[4], Mingjie Liu[1,3] & Lei Jiang[1,5]

Many biological organisms with exceptional freezing tolerance can resist the damages to cells from extra-/intracellular ice crystals and thus maintain their mechanical stability at subzero temperatures. Inspired by the freezing tolerance mechanisms found in nature, here we report a strategy of combining hydrophilic/oleophilic heteronetworks to produce self-adaptive, freeze-tolerant and mechanically stable organohydrogels. The organohydrogels can simultaneously use water and oil as a dispersion medium, and quickly switch between hydrogel- and organogel-like behaviours in response to the nature of the surrounding phase. Accordingly, their surfaces display unusual adaptive dual superlyophobic in oil/water system (that is, they are superhydrophobic under oil and superoleophobic under water). Moreover, the organogel component can inhibit the ice crystallization of the hydrogel component, thus enhancing the mechanical stability of organohydrogel over a wide temperature range ($-78$ to $80\,^\circ$C). The organohydrogels may have promising applications in complex and harsh environments.

[1] Key Laboratory of Bio-Inspired Smart Interfacial Science and Technology of Ministry of Education, School of Chemistry, Beihang University, No. 37 Xueyuan Road, Haidian District, Beijing 100191, China. [2] Beijing National Laboratory for Molecular Sciences (BNLMS), Key Laboratory of Green Printing, Institute of Chemistry, Chinese Academy of Sciences, Beijing 100190, China. [3] International Research Institute for Multidisciplinary Science, Beihang University, Beijing 100191, China. [4] National Center for Nanoscience and Technology, Beijing 100190, China. [5] Key Laboratory of Bio-inspired Materials and Interfacial Science, Technical Institute of Physics and Chemistry, Chinese Academy of Sciences, Beijing 100190, China. Correspondence and requests for materials should be addressed to M.L. (email: liumj@buaa.edu.cn).

In nature, many biological organisms, such as plants and animals living at high latitudes and altitudes, can survive subzero temperatures and resist the damages to cells from extra-/intracellular ice crystals[1,2]. This exceptional freezing tolerance originates from increased quantities of fatty species (for example, membrane lipids), which allow cell membranes to inhibit growth of ice and thus maintain their mechanical stability[3–5]. In biological systems, such dynamic coexistence of opposing components (for example, water and fatty molecules) is crucial to provide materials with complementary functionalities (for example, elasticity and freezing tolerance)[6,7]. This principle has inspired researchers to design and fabricate high-performance materials by combining two binary components with entirely opposite physiochemical properties at the nanoscale[8,9]. Examples include successful integration of electron donors and acceptors in semiconductors to improve solar cells[10]; the assembly of ions and ion-vacancy binary building blocks at the nanoscale to facilitate ion diffusion[11]; and alternating layers of strong and weak thermoelectric materials to enhance the thermoelectric efficiency[12]. We hypothesized that the binary cooperative complementary principle could also be applied in the design of hydrophilic/oleophilic heteronetwork gel materials (that is, organohydrogels) to overcome the weaknesses of individual hydrogels and organogels, and to produce a material that is highly adaptable in complex liquid environments[13–15].

In artificial materials, hydrogels are analogous to biological tissues in terms of their high water content and elastic properties[16–22]. However, conventional hydrogels composed of pure hydrophilic systems inevitably freeze and lose elasticity at subzero temperatures. Inspired by the freezing tolerance mechanisms found in nature, here we report a heteronetwork organohydrogel with stable elasticity over a wide temperature range ($-78$ to $80\,^{\circ}C$). The organohydrogel can be obtained through *in situ* polymerization of oleophilic polymer within a three-dimensional crosslinked hydrophilic matrix swollen with amphiphilic solvents. The interpenetrating heteronetworks allow the organohydrogel to adopt different surface configurations and network structures depending on the nature of surrounding phases. For instance, the organohydrogel is superoleophobic when equilibrated in water; behaviours that can be classified as hydrogel-like. However, when exposed to an oil-based environment, the organohydrogel became superhydrophobic, behaving like an organogel. Moreover, the organogel component can inhibit the ice crystallization in hydrogel, thus enhancing the mechanical stability of organohydrogel at subzero temperatures. This concept of complementary heteronetworks within a gel matrix will inspire researchers to design soft materials with complex and unusual functions.

## Results

**Preparation of heteronetwork organohydrogels.** To synthesize the organohydrogel, a hydrophilic polymer network (HPN) was first prepared by thermal polymerization of *N,N*-dimethyla-crylamide (DMA; 20 wt%) in the presence of methylene-bis-acrylamide (0.4 wt%) as a crosslinker (Fig. 1a). The preparation sequence of the organohydrogel is schematically shown in Fig. 1b. After adequately dehydrating the material with acetone, the HPNs were then swollen in an ethanol solution containing lauryl methacrylate (LMA; 34.1 wt%), *n*-butyl methacrylate (BMA; 35.0 wt%), ethylene glycol dimethacrylate (0.62 wt%) as a crosslinker and 2,2-diethoxyacetophenone (0.41 wt%) as a photoinitiator (Fig. 1a). Because ethanol has good affinity for both HPNs and oleophilic monomers, the solvent ensured effective infusion of the organogel precursors into HPN scaffolds. Finally, the HPNs swollen by ethanol solution were irradiated

under ultraviolet light to carry out *in situ* polymerization of the oleophilic polymer network (OPN) to obtain the organohydrogel featuring interpenetrating oleophilic/hydrophilic heteronetworks. The combination of different polymer networks in the organohydrogel results in a high tendency for the material to reconfigure its surface in response to the adjacent liquid phase. Consequently, the organohydrogel can switch its surface composition and related properties between that of hydrogels and organogels, or be locked in some intermediate state (Fig. 1c).

**Swelling and optical properties of organohydrogels.** The interpenetrating network structure of the organohydrogel spatially restricts the hydrophilic/oleophilic networks with respect to each other[23]. As a result of this effect, the volume of the organohydrogel hardly changes when it is equilibrated in water or oil, a stark contrast from the volume changes observed in homogeneous networks of organogels or hydrogels. For example, as shown in Fig. 2a, the HPNs were highly transparent and in a swollen state when equilibrated in water. However, when the same material was immersed in *n*-dodecane (oil), the HPNs stayed in a shrunken state. The OPNs showed the opposite effect (Fig. 2b), swelling in oil, but not in water. In contrast, the organohydrogel maintained a constant volume whether it was equilibrated in oil or water, when the weight ratio of OPN/HPN components in heteronetworks was $\sim$2:1 (Fig. 2c). Because the heteronetworks contain constituents with opposite solvent affinities, during equilibration one swollen polymer network is always constrained by the other shrunken polymer network, which further limits the collapse or excessive swelling behaviours of either network. As we increased the ratio of the OPN content in HPN scaffolds, the equilibrated swelling ratio ($Q$) of the organohydrogel in water obviously decreased, and the equilibrated volume difference of these samples in water and oil became smaller (Fig. 2d). When the OPN/HPN ratio reached 2.0–2.4, the organohydrogel maintained an almost constant volume when equilibrated in water or oil. The heteronetwork of the organohydrogel also induced a transparency change in response to solvent. When the OPN/HPN ratio reached 1:10, its transparency (wavelength 450–600 nm) decreased from $\sim$40.8% (homogeneous HPN) to $\sim$7.6% in water (Fig. 2e). When the OPN components were further increased, the organohydrogel became more opaque when immersed in water, which was probably caused by the scattering effect of aggregated OPN domains. However, when the same organohydrogel sample was equilibrated in pure oil, its transparency was much higher than that in water. For example, for the organohydrogel with an OPN/HPN ratio of 2:1, the transmittance was $\sim$35.5% in oil and $\sim$0.1% in water. Such transparency-responsive behaviours might be caused by the intrinsic network structures of the organohydrogel. During the preparation process, the HPNs serve as a scaffold to incorporate OPNs. When exposed to water, OPNs will aggregate, and water becomes separately distributed within the organohydrogel, causing visible light to scatter. In contrast, when exposed to oil, the OPN domains swell in the pores of the HPNs, thus restricting the aggregation of the HPNs. In this case, the light-scattering effect in oil became much weaker than that in water. Therefore, the organohydrogel with a higher OPN fraction possesses a higher transparency in oil than in water.

**Adaptive surface wettability and reconfigurable structures.** Owing to the opposite solvent affinities of HPNs and OPNs, the heteronetwork organohydrogel undergoes conformational rearrangement in selective solvents in a controlled and reversible manner[24,25]. Such structural reconfigurations switch the spatial distribution of the gelling liquid, which is trapped by the polymer

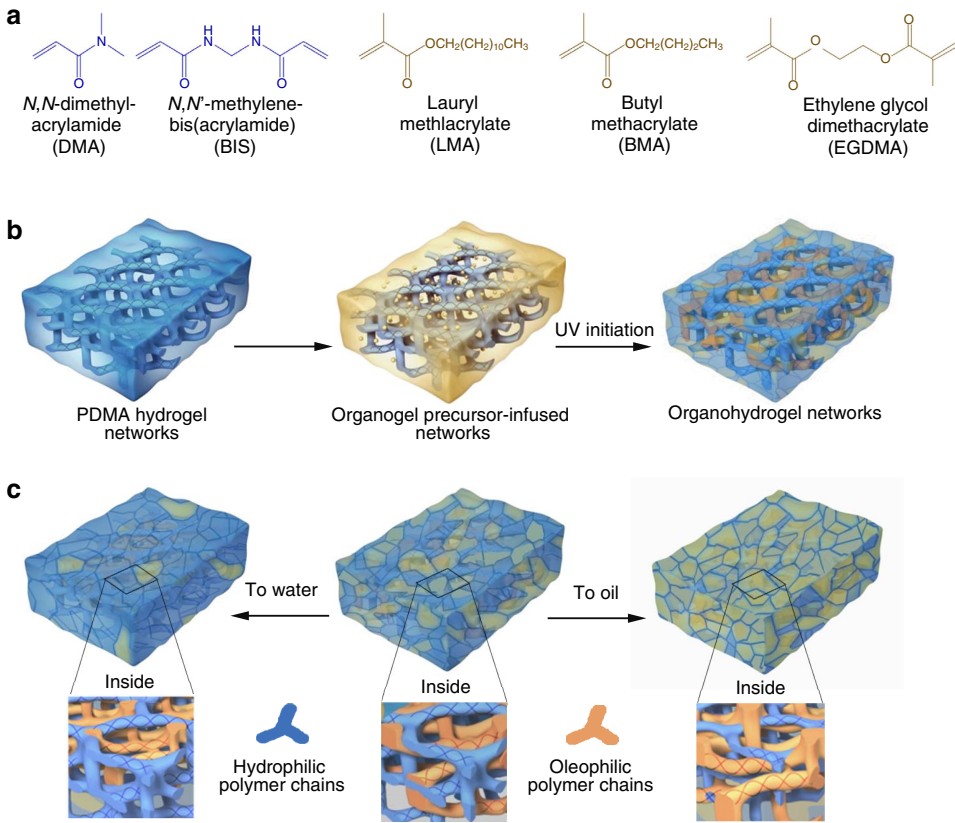

**Figure 1 | Schematic illustration of the preparation and reconfiguration of heteronetwork organohydrogels.** (**a**) The monomers and crosslinkers of the hydrogel (blue) and organogel precursors (yellow). (**b**) Preparation of the organohydrogel using an HPN as the scaffold to incorporate an OPN to form interpenetrating structures. (**c**) The organohydrogel reconfigures its surface in response to the liquid phase. In water, the HPNs can absorb water to make the organohydrogel perform like a hydrogel. Conversely, in oil, the OPNs can absorb the solvent to make the organohydrogel perform like an organogel.

networks; hence, the surface properties of these organohydrogels can be altered between hydrogel- and organogel-like states. For example, we observed that the surface of the organohydrogel became superoleophobic under water, featuring an oil contact angle (CA) of ~156° (Fig. 3a). In this case, the HPNs absorb water as soon as the organohydrogel is immersed in water. As the HPNs swell, the OPNs on the surface are gradually squeezed inwards, while the hydrogel component becomes dominant on the surface of the organohydrogel, which causes it to become superoleophobic. In contrast, when the organohydrogel was exposed to an oil environment, the oleophilic polymers appeared to reorient towards the oil phase and absorb oil, while the HPNs were collapsed inwards. In this scenario, the surface of the organohydrogel gradually became more organogel-like, and as a consequence became superhydrophobic as it was submerged in oil, featuring a water CA of ~154° (Fig. 3a).

Confocal Raman chemical imaging of organohydrogel surfaces equilibrated in different submerged environments revealed the surface component changes induced by the heteronetwork reconfiguration. As shown in Fig. 3b–d, the colours of images represent the characteristic peak area ratio of organogel $(2,777–3,091\,cm^{-1})$ to hydrogel $(3,163–3,583\,cm^{-1})$, which correlates qualitatively with the surface chemical composition. The colour transition from blue to red indicates the increase of organogel-like region due to much more oil participating in the networks equilibrium. From the Raman images we can also conclude that the spatial distribution of OPN and HPN domains on organohydrogel surface in submicroscale was homogeneous.

We carried out a systematic investigation to illustrate the relationship between the heteronetwork components and surface

adaptive wettability of the organohydrogel. As shown in Fig. 3e,a, a series of organohydrogel samples were immersed in water to reach an equilibrium state. As we increased the OPN/HPN ratio in the organohydrogel from 0.16 to 2.33, and equilibrated the material in n-dodecane, the water CA on the surface markedly increased, from ~10.7° to ~157.9°. This increase in water CA was caused by the increasing fraction of organogel microdomains formed on the organohydrogel surface under these conditions. In contrast, when we increased the OPN/HPN ratio, the oil CA of the organohydrogel as it was submerged under water slightly decreased, from ~163.8° to ~149.6°.

The CA changes differently for the two solvents because the swelling degree of the HPNs ($Q_{HPN}\sim$ 965%) in water is much larger than that of OPNs ($Q_{OPN}\sim$ 368%) in n-dodecane, which enables HPNs in water to form a high proportion of hydrogel domains on organohydrogel surface even when the ratio of OPN/HPN is ⩾1.0. When the OPN/HPN ratio was between ~1.50 and 2.03, the OPNs and HPNs generated optimal synergistic effects, allowing the organohydrogel surface to reversibly switch between superhydrophobic and superoleophobic in response to oil/water environments.

In the case of Fig. 3e, the optimized OPN/HPN ratio was achieved based on an HPN scaffold featuring a crosslinker concentration 0.4 wt%. Owing that crosslinking density has a large effect on the swelling degree of hydrogel networks, the optimized ratio of OPN/HPN to achieve adaptive surface switching between superhydrophobic and superoleophobic behaviours will in part depend on the crosslinking degree of the organohydrogel. Figure 3f shows a series of representative HPNs with different crosslinker concentrations and the

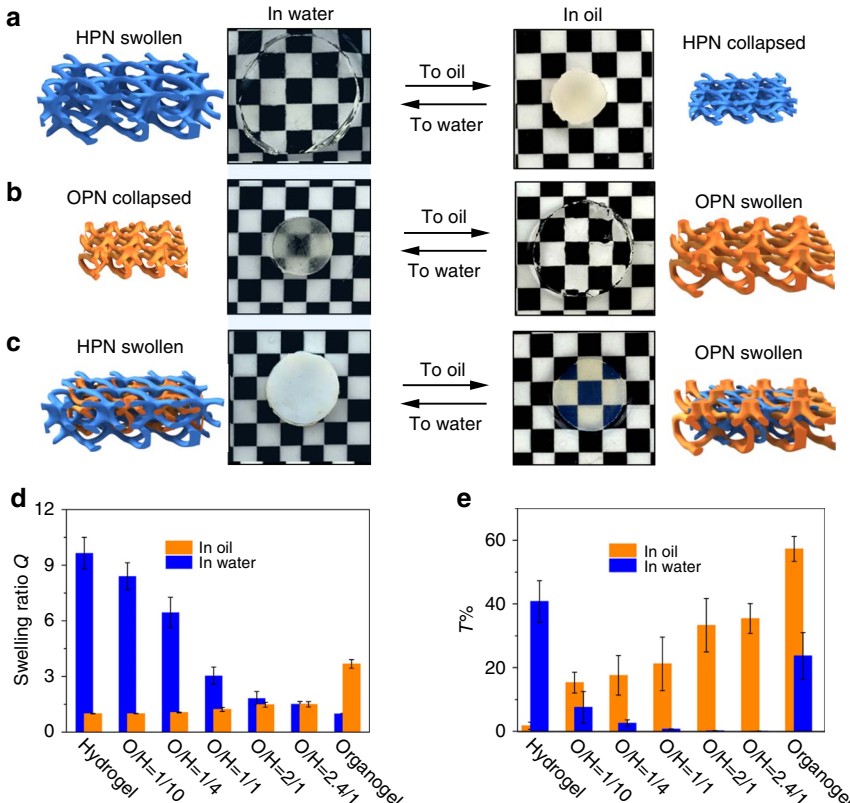

**Figure 2 | Swollen behaviours and transmittance change of gels with different network.** Optical images and corresponding schematic illustrations of (**a**) hydrogel, (**b**) organogel and (**c**) organohydrogel immersed in water and oil (that is, *n*-dodecane). The organohydrogel (OPN/HPN ∼ 2:1) was kept at a constant volume because of the interconstrained effect between the HPNs and OPNs. (**d**) The swelling ratio of the organohydrogels with different OPN/HPN ratios in water (blue) and oil (yellow) (O/H represents the weight ratio of OPN/HPN in the organohydrogels). (**e**) The corresponding transmittance (*T*%) of the organohydrogels when equilibrated in water (blue) and oil (yellow). The organohydrogels with higher OPN content and equilibrated in oil have a higher transparency than in water.

corresponding optimal OPN/HPN ratio that enables the organohydrogel to reversibly switch between superhydrophobic and superoleophobic in different liquid environments. As the crosslinker concentration increased, the swelling degree of the HPNs became gradually smaller, and accordingly a lower OPN fraction was necessary to keep the two networks in balance within the organohydrogel. For example, a hydrogel with a crosslinker concentration of 0.1 wt% required the optimal OPN/HPN ratio to be raised to ∼ 3.83, compared to a hydrogel with a crosslinker concentration 0.4 wt%, which required an optimal OPN/HPN ratio of just ∼ 2.03. The higher OPN fraction was necessary to constrain excessive swelling of the HPNs when the crosslinker concentration was lower.

The surface nature of the organohydrogel can adaptively switch between superhydrophobic and superoleophobic with high reversibility. As shown in Fig. 3g, the organohydrogel (OPN/HPN = 2.2:1) is dual superlyophobic when alternately equilibrated in water and oil, demonstrating CA ≥ 150° in all cases. In addition, this adaptive transition of the organohydrogel surface is fast, usually finishing in < 5 min, which we ascribe to that the interpenetrating network structures can accelerate the reorganization of the interface. The adaptive macroscopic wetting properties of the organohydrogels induced by surface reconfiguration suggest a promising application in smart gating systems, such as on-demand oil/water separation (Supplementary Figs 7 and 8). The principle goes that the interface equilibrium towards one phase can resist the traverse of the other incompatible phase, allowing only the compatible phase to pass through.

**Mechanical stability of organohydrogels.** In view of the bicontinuous water and oil phase embedded within the organohydrogel, we conjectured that such opposite solvents might play complementary roles on the material's elastic properties. In general, the solvent composition and content of a polymer gel are crucial in determining the gel's elasticity[26,27]. Owing to the freezing point of solvents, temperature will also inevitably influence elasticity. For example, hydrogels with water as a dispersion medium loses elasticity at subzero temperatures because the water freezes. The effects of temperature on the separate hydrogel and organogel components of the organohydrogel are shown in Fig. 4a–c. At low temperature, ranging from 0 to −20 °C, the storage modulus ($G'$) of the poly(DMA) hydrogel (90 wt% water) abruptly increased by almost $8 \times 10^3$ times (blue). At this temperature, the hydrogel turned into an ice-like solid that could be easily broken as it was bent (Fig. 4d). Alternatively, when the temperature was increased from 40 to 80 °C, $G'$ of the PDMA hydrogel gradually increased from 1,659.4 to 1,761.0 Pa. At these higher temperatures, hydrogen-bonding interactions between the PDMA network and water molecules become moderately weaker, causing the hydrogel to mildly dehydrate, resulting in a slightly higher $G'$[28]. For the organogel component in a dispersion medium of *n*-decane (12 wt%), the elastic properties remained stable at low temperature range (−20 to 0 °C) due to the low freezing point of *n*-decane (−29.7 °C). However, as the temperature was increased (20–80 °C), $G'$ of the organogel decreased by 41.2% (from 4,461 to 2,801 Pa) because of the thermal

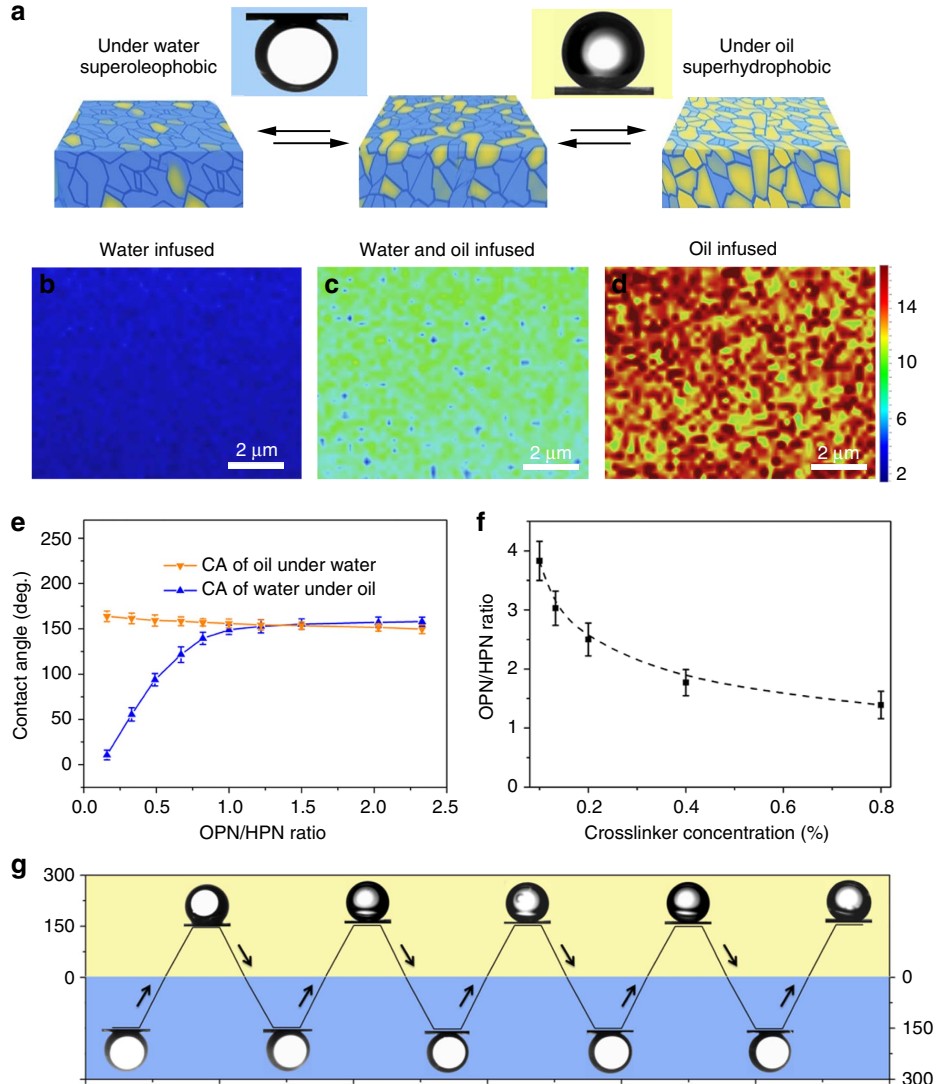

**Figure 3 | Adaptive surface wettability and reconfigurable structures of the organohydrogels. (a)** Reversible wetting transition of the organohydrogel surfaces submerged in water (superoleophobic) and in oil (superhydrophobic) by solvent-induced reconfiguration of the polymer networks. **(b–d)** Confocal Raman images showing the main components of surfaces of the organohydrogel equilibrated in **b** water alone, (**c**) both water and *n*-dodecane and (**d**) *n*-dodecane alone. (**e**) The effect of the OPN/HPN ratio on the CA of oil droplets on the sample equilibrated and submerged in water, and water droplets of the sample equilibrated and submerged in oil. When the OPN/HPN ratio was in the range of ∼1.50–2.03, the reversible superoleophobic and superhydrophobic transitions could be achieved. (**f**) The optimized OPN/HPN ratio for the HPN with different crosslinking densities to achieve the superoleophobic to superhydrophobic transition in water and oil, respectively. (**g**) The adaptive wetting transition shows good reversibility after alternately immersing the organohydrogel in water and oil.

plasticity of OPNs at a low crosslinking concentration 0.03% (Supplementary Fig. 14). Such a large reduction of $G'$ caused the organogel to turn from a freestanding to non-freestanding state (Fig. 4e).

In contrast, the mechanical strength of the heteronetwork organohydrogel was much higher than any single network hydrogel or organogel containing similar polymer concentrations (Supplementary Fig. 10a), demonstrating the enhanced mechanical strength of interpenetrating HPN/OPN double-network structures. More importantly, the organohydrogel showed stable elasticity over a wide temperature range. For organohydrogels containing 61.6 wt% water and 2.0 wt% *n*-decane, $G'$ only increased by 70.8% when the temperature decreased from 20 to −15 °C. Further increasing the weight per cent of *n*-decane caused the elasticity of the organohydrogels under subzero temperature to become more stable (Supplementary Fig. 12).

For example, $G'$ of the organohydrogel containing 38.5 wt% *n*-decane increased by only 3.9% at −15 °C compared with that at 20 °C. Apparently, the increase of $G'$ at subzero temperatures was due to the freezing of water inside HPNs. However, the OPN domains were still able to maintain the elasticity of organohydrogel at subzero temperatures. This may be attributed to the special ice crystallization in organohydrogels. The X-ray diffraction patterns of the ice crystals grown inside hydrogel and organohydrogel are shown in Fig. 4b, where the (1 0 0) diffraction peak of organohydrogels displays remarkable broadening and peak-splitting features. The broadening feature indicates the diminished grain size of ice or discrete crystallization forming in organohydrogels[29]. The peak splitting implies the lattice deformation of hexagonal ice in organohydrogel[30]. This difference indicates that the heterogeneous hydrophilic/oleophilic networks can greatly

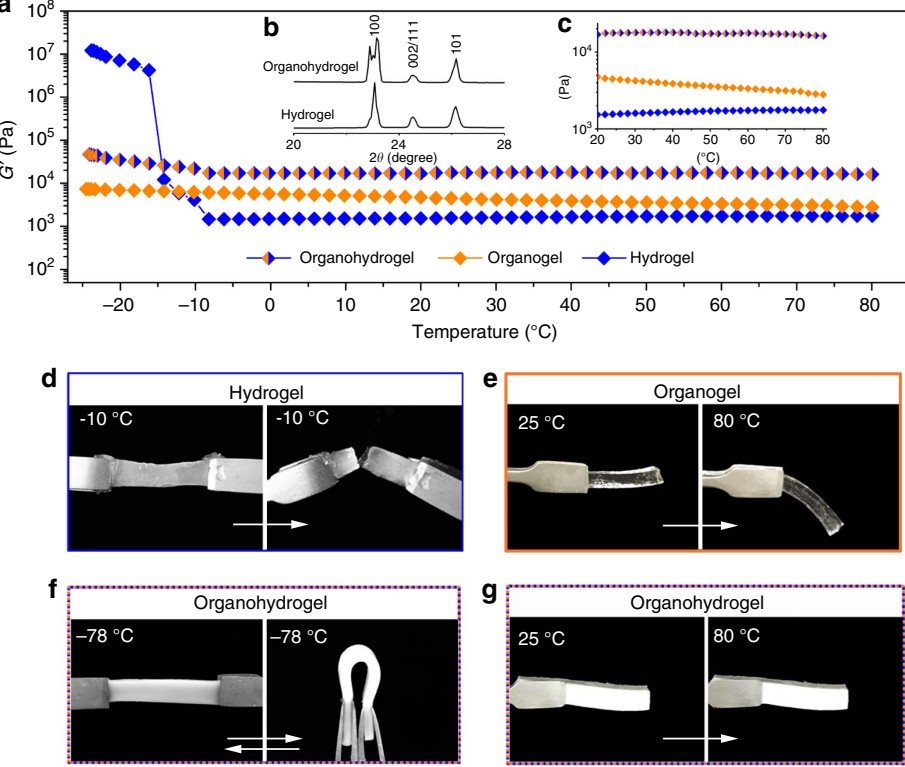

**Figure 4 | Elastic stability over a wide temperature range.** (**a**) $G'$ of the hydrogel, organogel and organohydrogel on a temperature sweep in the range of $-20$ to $80\,°C$ at a constant shear strain ($\gamma$) of $0.1\%$ and frequency ($\omega$) of $10\,rad\,s^{-1}$. (**b**) X-ray diffraction pattern of water frozen inside hydrogel and organohydrogel. (**c**) The magnified spectra of $G'$ of the hydrogel, organogel and organohydrogel versus temperature from 20 to $80\,°C$. (**d**) The hydrogel turned into an ice-like solid state that could easily be broken at $-10\,°C$. (**e**) The organogel turned into a non-self-standing state when heated to $80\,°C$. (**f,g**) The organohydrogel maintained almost stable elasticity at both (**f**) $-78\,°C$ and (**g**) $80\,°C$.

influence the water-to-ice transformation and further depress continuous ice generation. The polarization microscopy photographs in Supplementary Fig. 15 also reveal a remarkably contrast ice crystals inside hydrogel and organohydrogel accordingly. Large and continuous ice crystal domains could be found in frozen hydrogels, while no obvious ice crystal domains could be identified in frozen organohydrogel. This supports our conclusion that the ice crystallization in organohydrogels is greatly inhibited by the heterogeneous network. This mechanism is similar to the freezing tolerance mechanisms in nature. At subzero temperatures, antifreeze proteins with the heterogeneous hydrophobic/hydrophilic groups can lead to disordered structure of ice nucleation and further depress continuous ice generation[31]. In our system, owing to the discontinuous ice crystallizing in the polymer network, the organohydrogel can maintain its elasticity at subzero temperature.

The freezing tolerance of organohydrogels can be further enhanced by absorbing oil with a lower freezing point or reducing water content. For example, when we utilized heptanes as the equilibrating solvent of the OPNs, the organohydrogel maintained reversible bending and recovery of elasticity without fracture, even at temperatures as low as $-78\,°C$ (Fig. 4f). At a high temperature range, unlike the hydrogel increase in $G'$ and organogel decrease in $G'$, the organohydrogel maintained an almost constant $G'$ (Fig. 4c). As shown in Fig. 4g, compared with the organogel, the organohydrogel kept a freestanding state with no distortion at temperatures of $80\,°C$. We believe this stability originates from the complementary effect of HPNs and OPNs, in which the mild contraction of HPNs can compensate for the thermoplastic effect of OPNs at high temperature. Namely, the

OPNs enable the organohydrogel to remain soft at subzero temperatures and the HPNs ensure the hardness of the organohydrogel at higher temperatures.

## Discussion

In summary, taking inspiration from freezing tolerance organisms in nature, we prepared heteronetwork organohydrogels using crosslinked HPNs as scaffolds to incorporate oleophilic polymers. Owing to the reconfigurable HPN/OPN heteronetworks, the transparency and surface wettability of the organohydrogel could adapt to different liquid immersion environments, without significantly changing the gel volume. Importantly, the amphiphilic nature of the heteronetworks allows the organohydrogel to simultaneously contain water and oil as dispersion solvents in the three-dimensional network structures. The integration of OPNs and HPNs can compensate for the weaknesses of hydrogels and organogel, which guarantees that the organohydrogel will have stable elasticity over a wide temperature range. These organohydrogels seamlessly bridge hydrogels and organogels, and have promising applications in various fields, such as anti-icing, antiwaxing, antipainting and heterogeneous catalytic reactions.

## Methods

**Preparation of organohydrogel with interpenetrating heteronetworks.** To obtain organohydrogel with interpenetrating heteronetworks, dehydrated hydrogel scaffolds need be prepared firstly. PDMA hydrogel samples were prepared by thermal polymerization. The concentration and volume used in this paper are shown in Supplementary Table 1. The reaction was performed at $70\,°C$ for $3\,h$. The as-prepared hydrogel was dehydrated by acetone repeatedly. Second, the dehydrated hydrogel scaffolds were further immersed in organogel precursors in a cool dark place. The corresponding concentrations of organogel precursors used are

shown in Supplementary Table 2. In consideration of the mechanical property and oil-swollen capacity of organogel network, our oleophilic network was fabricated by copolymerization of two monomers, that is, LMA and BMA. In our case, LMA with long alkane side chains (12 C atoms) forms the soft and flexible PLMA network, which exhibits a low mechanical performance. In contrast, BMA with a short alkane side chain (four C atoms) usually forms the rigid and tough PBMA network, which can effectively enhance the mechanical strength of organogel network. Meanwhile, from the oil-swollen capacity perspective, PLMA network has a better oil-swollen capacity than the PBMA network. The LMA and BMA complement each other. We utilized the copolymer network of BMA and LMA monomers to improve both the mechanical performance and the oil-swollen capacity.

After the hydrogel scaffolds were fully infused with organogel precursors, they were irradiated under a high-pressure mercury arc lamp (Perfectlight 500 W) for ~60 min. The as-prepared organohydrogel was rinsed repeatedly with acetone to remove organic residues. Controlling the ratio of OPN and HPN was based on controlling the swelling degree of organogel precursors to HPN scaffolds.

**Optical transmission measurements.** Macroscopic optical images were all taken by using Canon EOS-60D. Optical transmission measurements for visible light were carried out using Ultraviolet–Visible Spectrophotometer (Shimadzu UV-2600), and the samples were tailored into the same size 20 mm × 10 mm × 4 mm. The measurement was carried out in air at room temperature after the polymer networks equilibrated completely in water/n-dodecane. The samples equilibrated with water were firstly processed with acetone to remove residual oil, and then repeatedly swollen with water for several times. Similarly, the samples equilibrated with oil (for example, n-dodecane) were processed with acetone to remove residual water, and then repeatedly swollen with oil (for example, n-dodecane) for several times.

**CA measurements.** All the CAs were measured using an OCA-20 machine (Dataphysics, Germany) at ambient temperature. The droplet volume was precisely controlled at 2 μl. More than five spots were taken per sample to obtain a mean value. The sample preparation in detail was follows: all samples including hydrogel, organogel and organohydrogel were cut into flat sheet with 2–4 mm thickness. All organohydrogel samples were firstly immersed in water to reach an equilibrated state (saturated state of swelling). Then, the data of CAs of oil (n-dodecane) on the samples under water were collected. Next, we needed to remove the free water on the organohydrogel samples with filter paper, and immerse the samples in oil (n-dodecane) to make the surface reach a stable state (based on many experimental experience, the CAs maintain stability after ~5 min of immersion in oil). The data of CAs of water on the samples under oil (n-dodecane) were collected.

**Raman spectroscopy and spatial Raman mapping.** Raman spectroscopy and spatial Raman mapping were performed using a Raman spectrometer (LabRAM HR Evolution; Horiba Scientific). The wavelength of the excitation laser was 532 nm. Raman maps (maximum scan range $\approx 8 \, \mu m \times 10 \, \mu m$) were collected using a spatial resolution of 200 nm. The collected spectra were processed using cosmic ray removal, noise filtering, baseline correction and normalization techniques based on the software of Lapspec-6. The imaging was developed by the Lapspec-6 software by calculating the ratio of characteristic peak area of organogel ($2{,}777$–$3{,}091 \, cm^{-1}$) to hydrogel ($3{,}163$–$3{,}583 \, cm^{-1}$).

**Rheological characterization.** Rheological characterization was carried out on a modular compact rheometer (MCR302; Anton Paar) with a parallel plate (15 mm) used to conduct oscillatory tests. The gel samples were cut into round plate with a diameter of 15 mm and a thickness of ~4 mm. Temperature-ramp experiments were carried out by cooling the samples at a rate of $3 \, °C \, min^{-1}$ from $20 \, °C$ to $-25 \, °C$, then keeping at $20 \, °C$ for 20 min at least and finallyheating samples at the same rate from 20 to $80 \, °C$, while monitoring the viscoelastic moduli of both the steps under small-amplitude oscillatory shear at an applied frequency of $\omega = 10 \, rad \, s^{-1}$ and strain amplitude of $\gamma_0 = 0.1\%$. Frequency sweeps at $25 \, °C$ temperatures were carried out over a range of $\omega = 0.1$–$20 \, rad \, s^{-1}$ at a strain amplitude of $\gamma_0 = 0.1\%$.

**Data availability.** The authors declare that the data supporting the findings of this study are available within the article and its Supplementary Information.

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

## Acknowledgements

This work was supported by the National Natural Science Foundation (21574004, 21421061), National Research Fund for Fundamental Key Projects (2013CB933000, 2012CB933800), the Key Research Program of the Chinese Academy of Sciences (KJZD-EW-M03), the Fundamental Research Funds for the Central Universities and the National 'Young Thousand Talents Program'.

## Author contributions

H.G., M.L. and L.J. designed the experiments. H.G., Z.Z. and M.L. analysed the data of experiments. Y.C. prepared dehydrated hydrogel samples. J.Z. conducted theoretical analysis on non-swellable phenomena. D.H., W.H., L.C., L.W. and J.Z. supported the characterization of samples. M.L. and H.G. wrote the manuscript.

**Additional information**

**Competing interests:** The authors declare no competing financial interests.

