## [Peer Review File · Nature Communications]

Reviewers' comments:

Reviewer #1 (Remarks to the Author):

Soft nanocomposites able to function over the large temperature range are needed for a myriad of applications, from advanced pharmaceuticals to smart materials (e.g., superoleophobic coatings for protective clothing). Inspired by physiological principles of natural freeze tolerance in biological systems, the authors describe an innovative synthesis of hetero-network organohydrogel with stable elasticity over a wide temperature range (-78 to 80°C). This is a well-written, elegant and thoughtful paper describing efforts to investigate the new platform for making soft nanocomposites. Experiments are well conducted, results are well documented and the manuscript reads very well. The conclusions are sound. The authors of the paper have clearly done a lot of work, and the results obtained by them are quite interesting.

The data are interesting in several regards. First they provide a careful measurement of the properties of organohydrogels and their behavior in different environment. Secondly, the most interesting part of these new organohydrogels is that they can switch the surface composition and some selected properties in response to the adjacent liquid phase that they are immersed in. Finally, investigations of the mechanisms of the freeze tolerance may provide a novel insight into many biomedical problems (e.g., development of protocols for cryopreservation and banking human organs, or preserving donor materials before transplantation, by immersing them in these organohydrogels?). Such studies are all too rare, thus, the data merit publication.

Reviewer #2 (Remarks to the Author):

The paper demonstrates freeze-tolerant organohydrogels with enhanced mechanical stability over a wide temperature of -78 to 80 °C. This work was inspired by exceptional freezing tolerance of plants and animals living in subzero temperatures. The gels are organohydrogels made of interpenetrating networks. The gels consisting of hydrophilic network (HPN) (DMA gel, 1st network) and oleophilic network (OPN) (LMA and BMA gel, 2nd network). The HPN swells in water and the OPN swells in oil. It also exhibits interesting surface properties depending on the surrounding phases; superoleophobic when equilibrated in water and superhydrophobic in oil-based environment. This is due to self-surface modification depending on the surface environment. Though each of these interesting properties can be expected from the chemical structure, i.e., interpenetrating networks of HPN and OPN, the authors find a new application to freeze-tolerant organohydrogels. It is an interesting piece of work and is worthy for publication in Nature Communication.

Before publication, the authors should answer the following questions and comments:

1) What is the reason why two components, i.e., LMA and BMA, were used for OPN? No explanation was given in the text. LMA is crystallizable component. Fig. S10 shows DSC curves taken for dried gels. No crystallization was detected. How about in wet condition, i.e., in water or in oil?

2) Inset of Fig. 4a: The XRD pattern of organohydrogel has a splitted 100 diffraction while that of hydrogel has a single peak. The difference is simply explained by the statement "the (100) diffraction peak of organohydrogels is much wider ..." (line 219 -). It does not look like the statement. Further explanation should be given.

Reviewer #3 (Remarks to the Author):

This paper by H. Gao et al. investigates the advanced material functions of this bio-inspired

organohydrogel and claims that such a hetero-network gel is of a stable elasticity over a wide temperature range. This is an interesting study and the gel material may have more advantage for low temperature applications than hydrogel. However, the material synthesis and characterization are lack of clarity which makes the innovation of the material of serious doubt. Specifically, there are several points given below.

1. In terms of freezing tolerance, the more detailed mechanism is needed. Have the authors compared with the other kinds of organogels? According to the description in this version, the freezing tolerance mechanism of gel seems to be only because the oil can withstand lower temperatures, in which, oil acts as a plasticizer to maintain the ductility of gels. If this trivial mechanism is the case, the scientific innovation of the work is limited and the stable elasticity is of doubt.

2. For mechanics of organohydrogel, the compression tests, as well as the unloading response are also required. Moreover, the viscoelastic property of the material is not very well presented. For instance, in Figure S8, I could not find the G'' data in the figure. For figure 4d, it is not clear if the deformed material can recover its shape after unloading. In addition to polymer concentration and water and/or oil content, have authors also checked the porosity and pore size of these gels? These two factors also have an influence on their mechanical properties.

3. More detailed characterization is required for the organohydrogel, such as porosity, the size and the uniformity of pores. It is now clear how the authors can determine the well entangled and crosslinked topology of the two polymer systems. And more detailed measurements are needed to support the suggested schematic illustration in Figure 1.

4. For the description of Figure 3a (in line 115-125), my question is what will happen if the organohydrogel is immersed into a sonicated water-oil mixture instead of immersing step by step? In addition, have authors measured the maximum water and oil content in an organohydrogel? If the organohydrogel is compressed, will water and oil be released or still bind well with the gel? These are important to understand the stability of the material.

5. For confocal Raman imaging, the authors also need to provide the optical image of the testing area, because the chemical image is also affected by the surface topographies of the samples. Moreover, the authors need to give more information about the Raman measurements, such as baseline correction etc. The single spectra in a specific pixel (such as a spectrum in green and black) region also need to be presented for comparison.

Referee 1

Comment:

Soft nanocomposites able to function over the large temperature range are needed for a myriad of applications, from advanced pharmaceuticals to smart materials (e.g., superoleophobic coatings for protective clothing). Inspired by physiological principles of natural freeze tolerance in biological systems, the authors describe an innovative synthesis of hetero-network organohydrogel with stable elasticity over a wide temperature range (-78 to 80°C). This is a well-written, elegant and thoughtful paper describing efforts to investigate the new platform for making soft nanocomposites. Experiments are well conducted, results are well documented and the manuscript reads very well. The conclusions are sound. The authors of the paper have clearly done a lot of work, and the results obtained by them are quite interesting.

The data are interesting in several regards. First they provide a careful measurement of the properties of organohydrogels and their behavior in different environment. Secondly, the most interesting part of these new organohydrogels is that they can switch the surface composition and some selected properties in response to the adjacent liquid phase that they are immersed in. Finally, investigations of the mechanisms of the freeze tolerance may provide a novel insight into many biomedical problems (e.g., development of protocols for cryopreservation and banking human organs, or preserving donor materials before transplantation, by immersing them in these organohydrogels?). Such studies are all too rare, thus, the data merit publication.

Response:

We appreciate this highly encouraging comment.

Answers to Comments of Reviewer 2

Comment:

The paper demonstrates freeze-tolerant organohydrogels with enhanced mechanical stability over a wide temperature of -78 to 80 °C. This work was inspired by exceptional freezing tolerance of plants and animals living in subzero temperatures. The gels are organohydrogels made of interpenetrating networks. The gels consisting of hydrophilic network (HPN) (DMA gel, 1st network) and oleophilic network (OPN) (LMA and BMA gel, 2nd network). The HPN swells in water and the OPN swells in oil. It also exhibits interesting surface properties depending on the surrounding phases; superoleophobic when equilibrated in water and superhydrophobic in oil-based environment. This is due to self-surface modification depending on the surface environment. Though each of these interesting properties can be expected from the chemical structure, i.e., interpenetrating networks of HPN and OPN, the authors find a new application to freeze-tolerant organohydrogels. It is an interesting piece of work and is worthy for publication in Nature Communication. Before publication, the authors should answer the following questions and comments.

Response:

We appreciate this highly encouraging comment.

Comment:

What is the reason why two components, i.e., LMA and BMA, were used for OPN? No explanation was given in the text.

Response:

We appreciate the reviewer's helpful question. In consideration of the mechanical property and oil-swollen capacity of organogel network, our oleophilic network was fabricated by copolymerization of two monomers, i.e., LMA and BMA. In our case, LMA with long alkane side chains (12 C-atoms) forms the soft and flexible PLMA network which exhibits a low mechanical performance. In contrast, BMA with a short alkane side chain (4 C-atoms) usually forms the rigid and tough PBMA network which can effectively enhance the mechanical strength of organogel network. Meanwhile, from the oil-swollen capacity perspective, PLMA network has a better oil-swollen capacity than the PBMA network. The LMA and BMA complement each other. We utilized the copolymer network of BMA and LMA monomers to improve both the mechanical performance and the oil-swollen capacity.

Comment:

LMA is crystallizable component. Fig. S10 shows DSC curves taken for dried gels. No crystallization was detected. How about in wet condition, i.e., in water or in oil?

Response:

We appreciate the reviewer's question. LMA is a crystallizable component, and the melting temperature is around -30°C to -40°C (refer to the black line in the following

image)¹. In Fig. S10 (Fig. S15 in the revised version) the DSC curves taken for dried gels were measured in the range of -30 °C to 200 °C, thus no crystallization melting peak of PLMA was detected. The DSC curves given below in temperature range of -70~-25°C display the crystallization (-45 °C) and melting peak (-35 °C) of PLMA for both dry organogel. In wet condition, i.e. in water or in oil, the crystallization/melting peaks of organogel do not show noticeable change in comparison with dry condition. We have followed reviewer’s suggestion and added more detailed explanation about DSC method and result analysis in the supplementary material.

Comment:

Inset of Fig. 4a: The XRD pattern of organohydrogel has a splitted 100 diffraction while that of hydrogel has a single peak. The difference is simply explained by the statement “the (100) diffraction peak of organohydrogels is much wider ...” (line 219 -). It does not look like the statement. Further explanation should be given.

Response:

We appreciate the reviewer’s helpful suggestion. In response to these comments, we have added the following explanation in the revised manuscript (Page 11).

“The XRD patterns of the ice crystals grown inside hydrogel and organohydrogel are shown in Fig. 4a (inset i), where the (100) diffraction peak of organohydrogels displays remarkable broadening and peak-splitting features. The broadening feature indicates the diminished grain size of ice or discrete crystallization forming in organohydrogels². The peak splitting implies the lattice deformation of hexagonal ice in organohydrogel³. This difference indicates the heterogeneous hydrophilic/oleophilic networks can greatly influence the water-to-ice transformation and further depress continuous ice generating. The polarization microscopy photographs in Supplementary Fig. S16 also reveal the remark contrast of ice crystals inside hydrogel and organohydrogel accordingly. Large and continuous ice crystal

domains could be identified in frozen hydrogels while no obvious ice crystal domains could be identified in frozen organohydrogel. This supports our conclusion that the ice crystallization in organohydrogels is greatly inhibited by the heterogeneous network. The described mechanism is similar to the freezing tolerance mechanisms in nature. At subzero temperatures, antifreeze proteins with heterogeneous hydrophobic/hydrophilic groups can lead to disordered structure of ice nucleation and further depress continuous ice generating⁴. In our system, owing to the discontinuous ice crystallizing in polymer network, the organohydrogel can maintain its elasticity at sub-zero temperature.”

Answers to Comments of Reviewer 3

Comment:

This paper by H. Gao et al. investigates the advanced material functions of this bio-inspired organohydrogel and claims that such a hetero-network gel is of a stable elasticity over a wide temperature range. This is an interesting study and the gel material may have more advantage for low temperature applications than hydrogel. However, the material synthesis and characterization are lack of clarity which makes the innovation of the material of serious doubt. Specifically, there are several points given below.

Response:

We appreciate this highly encouraging comment.

Comment:

In terms of freezing tolerance, the more detailed mechanism is needed. Have the authors compared with the other kinds of organogels? According to the description in this version, the freezing tolerance mechanism of gel seems to be only because the oil can withstand lower temperatures, in which, oil acts as a plasticizer to maintain the ductility of gels. If this trivial mechanism is the case, the scientific innovation of the work is limited and the stable elasticity is of doubt.

Response:

We appreciate this important comment. In organohydrogels, oil acting as a plasticizer is not the main reason to maintain the elasticity of gels at low temperatures. The most important mechanism is that the heterogeneous effect derived from hydrophilic and oleophilic networks can inhibit the formation of continuous ice crystals. This mechanism is similar to the freezing tolerance mechanisms in nature. At subzero temperatures, antifreeze proteins with the heterogeneous hydrophobic/hydrophilic groups can lead to disordered structure of ice nucleation and further depress continuous ice generating⁴. In our system, owing to the discontinuous ice crystallizing in polymer network, the organohydrogel can maintain its elasticity at sub-zero temperature. This mechanism can be further proved by the XRD and polarized optical microscopy measurement. In XRD, we see the broadening and peak-splitting features at the (100) diffraction peak for organohydrogels, which indicate the diminished grain size of ice and the lattice deformation of hexagonal ice, respectively²⁻³. The polarized optical images in Figure S16 further confirmed this result. To answer the reviewer's comments, we have added the mechanism explanation in the revised manuscript (Page 11).

“The XRD patterns of the ice crystals grown inside hydrogel and organohydrogel are shown in Fig. 4a (inset i), where the (100) diffraction peak of organohydrogels displays remarkable broadening and peak-splitting features. The broadening feature indicates the diminished grain size of ice or discrete crystallization forming in

organohydrogels². The peak splitting implies the lattice deformation of hexagonal ice in organohydrogel³. This difference indicates the heterogeneous hydrophilic/oleophilic networks can greatly influence the water-to-ice transformation and further depress continuous ice generating. The polarized optical microscopy photographs in Supplementary Fig. 16 also reveal the remark contrast of ice crystals inside hydrogel and organohydrogel accordingly. Large and continuous ice crystal domains could be found in frozen hydrogels while no obvious ice crystal domains could be found in frozen organohydrogel. Apparently, the ice crystallization in organohydrogels is greatly inhibited by the organogel network. This mechanism is similar to the freezing tolerance mechanisms in nature. At subzero temperatures, antifreeze proteins with the heterogeneous hydrophobic/hydrophilic groups can lead to disordered structure of ice nucleation and further depress continuous ice generating⁴. In our system, owing to the discontinuous ice crystallizing in polymer network, the organohydrogel can maintain its elasticity at sub-zero temperature.”

We also utilized other kind of olephilic polymer network, such as PDMS (Polydimethylsiloxane) network, to combine with hydrophilic polymer network. Similar freezing tolerance behavior was found.

Comment:

For mechanics of organohydrogel, the compression tests, as well as the unloading response are also required.

Response:

We appreciate the reviewer’s helpful suggestion. The loading-unloading compression tests of organohydrogels have been added to the revised supplementary material (Figure S12 in). From the compressive stress-strain curves, organohydrogel exhibits an increased compressive strength and elastic modulus in comparison to hydrogel and organogel. This demonstrates that the interpenetrating hetero-network structures of organohydrogels can greatly enhance the mechanical performance.

Figure S12. a, Stress-strain curves for organohydrogel (58 wt% water and 8 wt% n-decane), hydrogel (93 wt% water), and organogel (15 wt% n-decane) under uniaxial compression. The organohydrogel can sustain up to 190 kPa compression; while, the

hydrogel breaks at a stress of 54 kPa. **b-d**, the loading and unloading curves for the first cycle in compression, based on the samples of organohydrogel (58 wt% water and 8 wt% n-decane), hydrogel (93 wt% water), and organogel (15 wt% n-decane).

Comment:

Moreover, the viscoelastic property of the material is not very well presented. For instance, in Figure S8, I could not find the G'' data in the figure.

Response:

We have added the data of the loss storage G'' in the revised supplementary material (Figure S11 in Page 17).

Figure S11. **a**, Linear viscoelastic spectra (G' and G'') of the samples (hydrogel, organohydrogel and organogel) at 25 $^{\circ}\text{C}$. For organohydrogel with hetero-network, the G' at room temperature is higher than hydrogel or organogel with homogeneous polymer networks, indicating the enhanced effect of interpenetrating OPN/HPN structures. The three gels contained almost same concentrations of polymer networks. Hydrogel 71 wt% water; organohydrogel 69 wt% liquid mixture of water (54 wt%) and n-decane (15 wt%); organogel 67 wt% n-decane. **b**, the G'' of the samples (hydrogel, organohydrogel and organogel) from -20 to 80 $^{\circ}\text{C}$, which is corresponding with G' in Fig. 4a of the text.

Comment:

For figure 4d, it is not clear if the deformed material can recover its shape after unloading.

Response:

We appreciate the reviewer's helpful suggestion. Owing to the interpenetrating hetero-network structures, our organohydrogels can maintain its elasticity at subzero temperature. Following the reviewer's suggestion, we have performed the loading/unloading cycle to further prove this. As shown in Figure S14, the organohydrogel was compressed to a strain of 50 % at subzero temperature -78 $^{\circ}\text{C}$, and then in unloading process, the deformed sample easily recovered its original shape.

Figure S14. The loading and unloading curves for the first cycle in compression, based on the samples of organohydrogel (~18% n-heptane and ~10% water).

Comment:

In addition to polymer concentration and water and/or oil content, have authors also checked the porosity and pore size of these gels? These two factors also have an influence on their mechanical properties.

More detailed characterization is required for the organohydrogel, such as porosity, the size and the uniformity of pores. It is now clear how the authors can determine the well entangled and crosslinked topology of the two polymer systems. And more detailed measurements are needed to support the suggested schematic illustration in Figure 1.

Response:

We appreciate this important comment. Besides of Raman and AFM characterization, the scanning electron microscopy (SEM) has been performed on hydrogel and organohydrogel surfaces to reveal the detailed information about porosity, the size and the uniformity of pores. (Figure S6 in Page 11).

From the SEM images, we see that the sizes of the pores of organohydrogel (c-d) are much smaller than those of hydrogel (a-b), which is due to the filling effect of oleophilic networks. Meanwhile, the more uniform and compacted pore structures of organohydrogel enables stronger mechanical properties in comparison to hydrogel.

Figure S6. The scanning electron microscopy images show the morphologies and porosity of hydrogel (a-b) and organohydrogel (c-d). Both of them have evident porous structures. The pores on hydrogel are most at the micrometer scale. In contrast, compacted pores are observed on organohydrogel. The pores on organohydrogel are more uniform and most at several hundred nanometer scale, because of the filling effect of oleophilic networks.

Comment:

For the description of Figure 3a (in line 115-125), my question is what will happen if the organohydrogel is immersed into a sonicated water-oil mixture instead of immersing step by step?

Response 7:

Following the reviewer's suggestion, we have performed the experiments in which organohydrogel was immersed into different types of sonicated water-oil mixtures (i.e. water-in-oil emulsion and oil-in-water emulsion, Figure S7 in Page 12). When the organohydrogel is immersed into the oil-in-water emulsion, the surface becomes hydrophilic. This is because the hydrophilic network is preferentially swollen by the continuous aqueous phase in oil-in-water emulsion. Conversely, when organohydrogel is immersed into water-in-oil emulsion, its surface becomes hydrophobic. It indicates that the oleophilic networks are preferentially swollen by the continuous oil phase. Therefore, the surface property of our organohydrogel is mainly determined by the continuous phase of the emulsion, and further interaction between internal phase of the emulsion and the corresponding polymer network is prohibited by the surface.

Figure S7. The organohydrogel with reversible surface transformation was immersed in sonicated oil-in-water and water-in-oil emulsions, respectively. The surface turned hydrophilic after immersed in oil-in-water emulsions, because water as the continuous phase makes the hydrophilic networks swollen preferentially, and thus the surface turns hydrophilic CA~62.5°. In contrast, the surface turned hydrophobic after immersed in water-in-oil emulsions, because oil as the continuous phase makes the oleophilic networks swollen firstly, and thus the surface turn to be hydrophobic CA~106.5°.

Comment 8:

In addition, have authors measured the maximum water and oil content in an organohydrogel?

Response 8:

The maximum water and oil content of organohydrogel is related to the crosslinking degree of hydrophilic and oleophilic networks, and the ratio of hydrophilic and oleophilic networks. The maximum water and oil content of organohydrogel used in our system are approximately 80 wt% and 38 wt%, respectively.

Comment 9:

If the organohydrogel is compressed, will water and oil be released or still bind well with the gel? These are important to understand the stability of the material.

Response 9:

In our system, the water and oil cannot be released from the organohydrogel under compression. This indicates that water and oil are bound well with the hydrophilic and oleophilic networks.

Comment 10:

For confocal Raman imaging, the authors also need to provide the optical image of the testing area, because the chemical image is also affected by the surface topographies of the samples.

Moreover, the authors need to give more information about the Raman measurements, such as baseline correction etc.

The single spectra in a specific pixel (such as a spectrum in green and black) region also need to be presented for comparison.

Response 10:

We appreciate this important comment. We have added the detailed information and corresponding data about the Raman imaging in the revised supplementary material.

The optical images of the testing area have been given in Figure S4. d-f. The optical images of the testing positions show the flat surface topographies at micrometer scale. And the detailed information about the Raman measurements has been added in the revised supplementary material Page 4.

“Raman spectroscopy and spatial Raman mapping were performed using a Raman spectrometer (LabRAM HR Evolution, Horiba Scientific). The wavelength of the excitation laser was 532 nm. Raman maps (maximum scan range $\approx 8 \mu\text{m} \times 10 \mu\text{m}$) were collected with a spatial resolution of 200 nm. The collected spectra were processed using cosmic ray removal, noise filtering, baseline correction, and normalization techniques based on the software of Lapspec-6. The imaging was developed by Lapspec-6 software via calculating the ratio of characteristic peak area of organogel ($2777\text{-}3091 \text{ cm}^{-1}$) to hydrogel ($3163\text{-}3583 \text{ cm}^{-1}$).”

From the comparison, we can see that the pixel regions with different colors display distinct single spectra, especially in $3163\text{-}3583 \text{ cm}^{-1}$ range related to hydroxyl groups stretching from water. The organohydrogel sample pretreated in water (g- i and h) shows the strong peak in $3163\text{-}3583 \text{ cm}^{-1}$ range, which indicates that the organohydrogel pretreated with water has more hydrophilic hydroxyl polar groups on the surface. However, the organohydrogel sample pretreated in n-dodecane (g-i i and i) has no apparent peak in the same range due to few hydrophilic hydroxyl polar groups on its surface.

Figure S4. Raman information of individual organohydrogel equilibrated with pure water (the left column), water and oil (the middle column), and pure oil (the right column). **(a-c)** Raman spectra of all scanning points ($\approx 2,000$ points) in scanning range ($8 \mu\text{m} \times 10 \mu\text{m}$). **(d-f)** The optical images of the scanning position. In micro-scale, all of the three surfaces are basically flat. **(g)** The points with different color are marked in the three images (**i-iii**); and the corresponding spectra are shown in **(h-j)**.

Reference

1. Xu, Y. *et al.* Double-grafted cylindrical brushes: synthesis and characterization of poly(lauryl methacrylate) brushes. *Macromol. Chem. Phys.* **208**, 1666–1675 (2007).
2. Ehre, D. *et al.* Water freezes differently on positively and negatively charged surfaces of piezoelectric materials. *Science* **327**, 672–675 (2010).
3. Varshney, D. *et al.* Synchrotron X-ray diffraction investigation of the anomalous behavior of ice during freezing of aqueous systems. *J. Phys. Chem. B* **113**, 6177–6182 (2009).
4. Liu, K. *et al.* Janus effect of antifreeze proteins on ice nucleation. *Proc. Nat. Acad. Sci. USA.* **113**, 14739–14744 (2016).

REVIEWERS' COMMENTS:

Reviewer #2 (Remarks to the Author):

The authors answered my questions and revised the manuscript. I believe that the manuscript is now suitable for publication in Nature Communication.

Reviewer #3 (Remarks to the Author):

I have carefully gone through the revised paper and the response letter and I am quietly satisfied as the authors have addressed all my former questions by conducting new tests and discussions. I have no other problem for its acceptance for publication in Nature Communications.

REVIEWERS' COMMENTS:

Reviewer #2 (Remarks to the Author):

The authors answered my questions and revised the manuscript. I believe that the manuscript is now suitable for publication in Nature Communication.

Reviewer #3 (Remarks to the Author):

I have carefully gone through the revised paper and the response letter and I am quietly satisfied as the authors have addressed all my former questions by conducting new tests and discussions. I have no other problem for its acceptance for publication in Nature Communications.

Response: We thank both reviewers for recommending our work to be published without further revisions.